# Open-label placebo treatment of women with premenstrual syndrome: study protocol of a randomised controlled trial

Antje Frey Nascimento ![ORCID],[1] Jens Gaab,[1] Irving Kirsch,[2] Joe Kossowsky,[1,2,3] Andrea Meyer,[4] Cosima Locher ![ORCID] [1,5]

[1]Division of Clinical Psychology and Psychotherapy, Department of Psychology, University of Basel, Basel, Switzerland
[2]Program in Placebo Studies, Beth Israel Deaconess Medical Center, Harvard Medical School, Boston, Massachusetts, USA
[3]Department of Anesthesiology, Critical Care & Pain Medicine, Boston Children's Hospital, Harvard Medical School, Boston, Massachusetts, USA
[4]Division of Clinical Psychology and Epidemiology, Department of Psychology, University of Basel, Basel, Switzerland
[5]School of Psychology, University of Plymouth, Plymouth, UK

**Correspondence to**
Antje Frey Nascimento;
antje.freynascimento@unibas.ch

## ABSTRACT

**Introduction** Recent evidence suggests that for certain clinical conditions, placebos can improve clinical outcomes even without deception. These so-called *open-label placebos* (OLPs) bear the advantage of a significant lower risk of adverse events and comply with ethical principles. Although premenstrual syndrome (PMS) seems to be considerably susceptible to placebo effects, no study has examined open-OLP responses on PMS.

**Methods and analysis** To test the efficacy of OLPs in women suffering from PMS, a clinical randomised controlled trial including two OLP study groups (with and without treatment rationale) was designed to investigate on the effect on PMS. PMS symptoms are monitored on a daily basis via a symptom diary, adverse events are monitored intermittently. The study started in spring 2018 and patients will be included until a maximum of 150 participants are randomised. Besides the primary outcome PMS symptom intensity and interference, an array of further variables is assessed. Multilevel modelling will be used for data analyses.

**Ethics and dissemination** Ethics approval was obtained from the Ethics Committee Northwest and Central Switzerland. Results of the main analysis and of secondary analyses will be submitted for publication in peer-reviewed journals.

**Trial registration numbers** (1) ClinicalTrials. gov (NCT03547661); (2) Swiss national registration (SNCTP000002809).

## Strengths and limitations of this study

► This is the first clinical randomised controlled trial to test an open-label placebo (OLP) intervention in premenstrual syndrome (PMS).
► Moreover, this trial focuses on OLP responses on a somatic and a psychological outcome.
► To tackle the challenge of variations in PMS complaints and menstrual cycle lengths within subjects, enrolled women are comprehensively and prospectively screened for PMS and the primary outcomes are also prospectively assessed.
► We test the effect of a plausible, comprehensive OLP treatment rationale, which may be relevant for a better understanding of effects of treatment rationales in general.
► Since different definitions of PMS exist, we based our definition on the most highly recommended diagnosis assessment criteria and assessment tools.

response.[6] To date, no study has examined OLP responses on premenstrual syndrome (PMS), although PMS appears to be susceptible to placebo effects.[7–9] Moreover, a further analysis of the effect of the OLP treatment rationale can inform about intervention-independent effects of treatment rationales. Hence, we set out to examine OLP responses in PMS and the relevance of a plausible treatment rationale.

## RATIONALE

PMS entails clinically significant somatic and psychological symptoms in the premenstrual phase of the menstrual cycle, causing substantial distress and functional impairment.[10] Worldwide, a considerable amount of women of reproductive age are affected with a pooled prevalence of 47.8%.[11] Subsequently, a myriad of distinctive therapies are prescribed for PMS—including pharmacological and phytopharmaceutic drugs as well as complementary interventions[12]—yet there is limited evidence for some interventions (eg,

## INTRODUCTION

Recent evidence suggests that in certain clinical conditions—such as chronic low-back pain,[1] irritable bowel syndrome,[2] rhinitis,[3] and cancer-related fatigue[4]—placebos improve clinical outcomes even without deception. These so-called open-label placebos (OLPs) are prescribed with a plausible scientific treatment rationale, encompassing that placebos are powerful; the body automatically responds to the intake of placebos; a positive attitude towards the intervention can be helpful; and taking the placebo pills faithfully is critical.[1 2 5] We have shown in a standardised heat pain experiment that the provided treatment rationale is crucial for the elicitation of an OLP

Chaste tree; anxiolytics) while other empirically validated interventions (eg, selective serotonin reuptake inhibitors, hormonal therapy) show considerable side-effects.[13]

Notably, PMS seems to be highly susceptible to placebo effects: The Royal College of Obstetricians and Gynaecologists alerts to substantial placebo responses in randomised-controlled PMS intervention trials of 36%–43%[7] and different studies reported considerable placebo effects on PMS.[8 9 14] Furthermore, PMS offers the opportunity to investigate OLP effects on *both* somatic and psychological symptoms since symptom diaries like the German PMS symptom diary inquire about physical (eg, breast tenderness, headaches, joint and muscle pain) as well as psychological symptoms (eg, depressed mood, hopelessness, irritability). In addition, we are the first to assess the effect of a comprehensive OLP treatment rationale in a clinical population. We hypothesise that first, women obtaining an OLP treatment with a treatment rationale will report less PMS symptom intensity and interference during the course of the intervention and at follow-up, compared to women obtaining an OLP treatment without a treatment rationale. Second, we assume that women receiving an OLP treatment without a treatment rationale will also show a higher decrease in PMS symptom intensity and interference in comparison to the treatment as usual (TAU) group which is not given any of the treatment-specific information that the OLP groups receive nor the rationale for the OLP treatment.

## OBJECTIVES
### Primary objectives
The primary objective of this study is to test the effectiveness of an OLP treatment in women suffering from moderate to severe PMS in regard to PMS symptom intensity and interference. Also, we examine whether there is a group difference across time. Moreover, we test whether a comprehensive OLP treatment rationale has an effect on the OLP response compared to omitting the treatment rationale.

### Secondary objectives
1. To test whether an OLP response is observed for somatic or on psychological symptoms.
2. To assess the effect of the OLP treatment on PMS impact.
3. To evaluate the effect of the OLP intervention on quality of life.
4. To assess whether there is any impact of elevated baseline anxiety or depression levels on the OLP response.
5. To investigate whether there is any effect of the OLP intervention on relationship satisfaction.
6. To assess whether positive attitudes towards complementary medicine are associated with OLP responses.
7. To assess the impact of attitudes towards placebos on the OLP response.

---

> ### Box 1 Overview about research questions and study design
>
> **Research questions:**
> ► Does an open-label placebo (OLP) intervention alleviate premenstrual syndrome (PMS) symptoms over time?
> ► Do participants who receive a comprehensive OLP treatment rationale show a greater OLP response in comparison to participants who do not receive any treatment rationale over time?
>
> **Population:**
> Women suffering from a moderate to severe PMS, aged between 18 and 45 with a regular menstrual cycle and no essential comorbidities.
>
> **Intervention groups:**
> There are two intervention groups: one OLP with a plausible treatment rationale (OLP+) and an OLP without a plausible treatment rationale (OLP−) group. Both groups obtain openly administered placebo pills, hence two pills per day for 6 weeks.
>
> **Comparison:**
> A treatment as usual group without any intervention but the same number of study contacts as the intervention groups and the same study procedures serves as control group.
>
> **Outcome:**
> Primary endpoint are PMS symptom intensity and interference measured by means of a PMS symptom diary.
>
> **Time of study duration:**
> Approximately 2 years.

---

8. To evaluate whether the appraisal of the treatment provider has any effect in regard of the OLP effect.
9. To evaluate the intervention credibility in the two intervention groups.
10. To investigate whether expectancy of relief and desire for relief have an effect on the OLP response.
11. To assess treatment adherence of placebo pill intake.

## METHODS AND ANALYSIS
### Study design
A single-centre, randomised controlled clinical trial of an OLP intervention on women with PMS using a parallel group between-subject design with three study groups: a TAU group; an OLP with a plausible treatment rationale (OLP+) and an OLP without a treatment rationale (OLP−) group. The first participant was enrolled and randomised in August 2018, and the study is expected to be concluded by spring 2020 with the planned inclusion of 150 study participants. The study is being conducted at the Division of Clinical Psychology and Psychotherapy at the Faculty for Psychology of the University of Basel. For an overview, please see box 1. The design of the trial is summarised in figure 1 and all steps and aspects are delineated below (see also table 1 for an overview of the schedule).

### Subjects
In total, 150 women suffering from moderate to severe PMS will be randomly allocated to one of the three study groups.

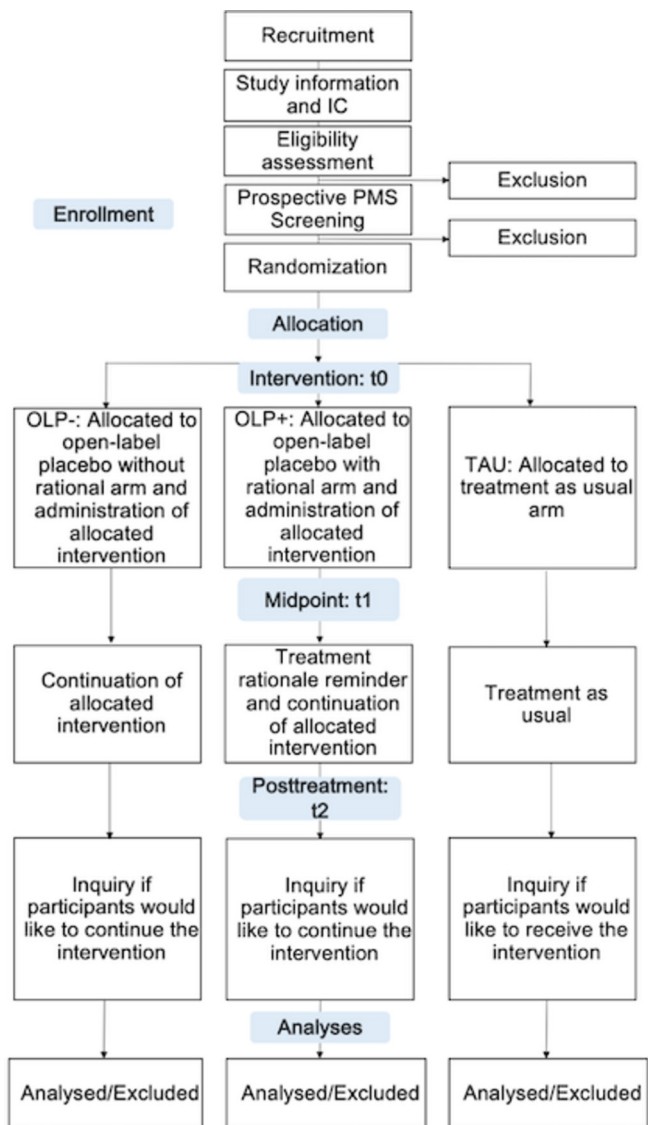

**Figure 1** Study design and flow of participants. IC, informed consent; OLP, open-label placebo; PMS, premenstrual syndrome; TAU, treatment as usual.

### Patient and public involvement statement

Previous studies about patient experiences in OLP and placebo trials of irritable bowel syndrome were consulted for this study to include patient experiences.[15]

### Patient eligibility

Women aged between 18 and 45 years with a regular menstrual cycle, suffering from moderate to severe PMS reported by prospective self-report are eligible for study participation.[16] A moderate to severe PMS diagnosis and the primary outcomes are identically assessed and calculated (for more details, see the Primary outcome measures section). Moreover, women have to be capable of consent, fluent in German and have a menstrual cycle duration ranging from 24 to 31 days. The severity of PMS must warrant the treatment of at least one functionally impairing physical or psychological symptom based on the women's self-report.[10] Also, participants have to know a general

practitioner or gynaecologist who they would consult if necessary. The recent initiation of a new medication (ie, within the last 30 days) leads to study exclusion or delayed enrolment, whereas the intake of any psychopharmacological or psychotropic substance or alcohol abuse lead to exclusion. Further exclusion criteria are pregnancy, breast feeding, indications of a psychiatric or gynaecological disorder, suicidality, further drug intake affecting PMS symptoms, a body mass index above 30, hypersensitivity or allergy to the placebo pills, participation in another study with drugs or in another PMS study within the last 3 months and participation in psychotherapy due to PMS.

### Recruitment

Participants are recruited via advertisement for 'a novel efficacy study of an integrative and side-effect free intervention against premenstrual complaints' with flyers and posters at ambulant gynaecologists in the Region of Basel and in the gynaecological clinic of the University Hospital of Basel. Moreover, different internet platforms are used for advertisement and different public facilities and stores around Basel, including the German border region.

Potential participants are informed that participation in the study is voluntary, withdrawal of consent is possible at any time without mentioning reasons, and refusal of study participation will not lead to any negative consequences. Interested women obtain a study information form (online supplementary materials 1 and 2) and all relevant information are also provided during the informed consent process with the opportunity to clarify questions. Women are only enrolled after written informed consent obtainment.

### Trial treatment and study arms

Following recruitment and verification of study eligibility - including a prospective PMS screening for one menstrual cycle - qualified women are allocated randomly by means of a random allocation sequence using the built-in random number generator in Microsoft Excel to one of the three study groups at the study visit (t0) and also are informed before any intervention, respectively the control contact starts: (1) a TAU group, (2) an OLP+ group and (3) an OLP– group (see also figure 2).

1. The TAU group serves as a control group and has the same amount of contacts with the study team. However, participants of this group do not receive any intervention nor any information concerning the intervention. However, they are told why the control group is valuable and essential for the trial and that they will receive the intervention after study conclusion if desired. They are allowed to continue any medication intake, if the substance does not lead to study exclusion and PMS complaints are still prevalent. The same is warrant for both intervention groups.

2. The OLP+ group obtains the intervention with a plausible treatment rationale. The intervention is provided at the personal meeting at the study site (t0), followed by a second study contact after 3 weeks by phone or if desired by participants again at the study site (t1) and a

**Table 1** Study schedule

| Study periods | Screening phase | | Intervention phase | | | | | Post-treatment study contact & OA (t2) |
|---|---|---|---|---|---|---|---|---|
| | Preliminary screening of eligibility | Prospective PMS screening | L1 OA | Personal meeting (t0) | L2 OA | Midpoint study contact | L3 OA | |
| Visit/contact | Phone/post/online | At home (diary form) | Online | At study site | Online | At study site or phone/Skype | Online | Phone +post-treatment online |
| Day of menstrual cycle (MC) | −31−−1 day of 1. MC | 1. day of 1. MC until end of 1. MC | −5−−1 of 2. MC | 1–14 day of 2. MC | −5−−1 of 3. MC | 1–14 day of 3. MC | −5−−1 of 4. MC | 1–14 day of 4. MC |
| Menstrual phase | At any point | One whole MC | Luteal | Follicular | Luteal | Follicular | Luteal | Follicular |
| Time (min) | 20 | 2 daily for 1 MC | 10 | 60 | 10 | 15 | 10 | 30 |
| Study information/informed consent | x | | | | | | | |
| Inclusion/exclusion criteria | x | x | | | | | | |
| Randomisation | | | | x | | | | |
| Placebo dragée intake in intervention arms | | | | x | x | x | x | |
| Narrative for OLP+ group | | | | x | | x | | |
| **Primary outcome** | | | | | | | | |
| PMS symptom diary | | x | x | x | x | x | x | |
| **Safety variables** | | | x | x | x | x | x | x |
| Suicidality assessment | x | | | x | | x | | x |
| Side-effects assessment | | | | | | x | x | x |

L, luteal phase; OA, online assessment; OLP, open-label placebo; PMS, premenstrual syndrome.

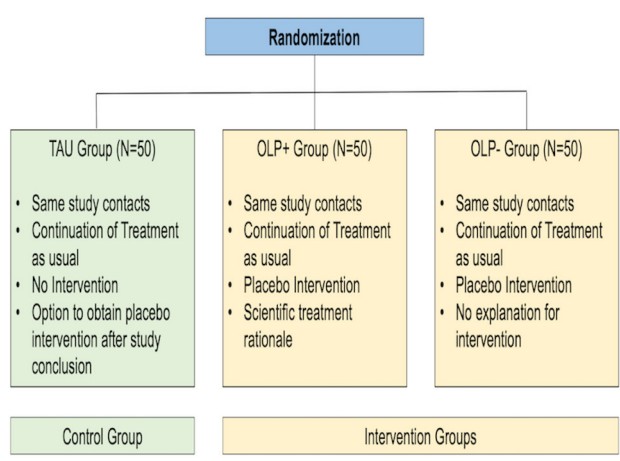

**Figure 2** Overview of study groups. OLP, open-label placebo; TAU, treatment as usual.

last post-treatment phone contact and assessment after the second treatment cycle (t2). The intervention consists of an administration of OLP pills including the instruction to take two pills a day for 6 weeks. The pill intake in both OLP intervention groups starts at the very day after the study visit (t0). The study visit (t0) occurs during the first 14 days of the menstrual cycle of each participant, that is, in the follicular phase. Due to the cyclicity of PMS symptoms, we decided to provide the intervention for a longer period of time in contrast to previous OLP studies[1–3] to encompass at least one whole menstrual cycle. After the explicit mentioning of the pharmacological inertness of the pills in both groups, the OLP+ group receives—in contrast to the OLP– group—additionally a comprehensive OLP treatment rationale (see online supplementary material 3) at t0 as well as at t1. The discussion points encompass: (a) that placebos can be powerful, also for PMS, (b) an explanation on why placebos work, (c) that faithful pill intake is crucial and that doubts do not have a negative effect on the treatment, (d) that placebos have also been shown to work without deception. Moreover, a video sequence of about 2 min is shown of an American NBC news report with German subtitle about an OLP intervention on irritable bowel syndrome including a patient report (see https://www.youtube.com/watch?v=uv0SuWKZjsI&t=1s). Neither participants nor study team members are blinded. Yet besides the treatment provider, study team members are not informed about group allocation to minimise investigator bias. As investigational product, pink oral placebo pills—'P-Dragees rosa Lichtenstein' of the German brand *Winthrop*—are administered which are certified placebo pills without any active ingredient and with validated quality. Each pill contains the following substances: lactose monohydrate; magnesium stearate (Ph. Eur.); microcrystalline cellulose; highly dispersed silicon dioxide; white clay, magrogol glycerolhydroxy stearate (Ph. Eur.); Arabic gum; montanglycol wax; povidone (K 25); talcum; titanium dioxide (E 171); erythrosine; aluminium salt (E

127); calcium carbonate; sucrose; glucose syrup; maize starch; macrogol 6000. Also, participants of both intervention groups receive a certificate, confirming that the study medication only contains placebo pills.

3. The OLP– group obtains the same intervention and intake instruction like the OLP+ group and the proceeding is identically. However, no treatment rationale or further treatment explanation is provided.

### Detailed screening and PMS assessment
After informed consent obtainment, participants are comprehensively screened for study eligibility. Therefore, the following information is collected online:
- Verification of eligibility according to eligibility criteria.
- Medical and gynaecological history.
- Assessment of sociodemographic data.
- Assessment of PMS retrospectively.

In a second step, women fulfilling eligibility criteria and PMS criteria retrospectively, start a prospective PMS screening with daily online surveys as part of a PMS symptom diary assessment for one menstrual cycle for the screening and in total for three menstrual cycle for primary outcome assessment (see below).

## OUTCOME MEASURES: PRIMARY, SECONDARY AND TERTIARY
### Primary outcome measures
The German PMS symptom diary consists of 27 symptoms and of three further items assessing interference in different domains (ie, work or school; social activities; relations with others). Usually, a four point Likert scale is employed, however, in line with Endicott *et al*[17] we apply a six point Likert scale to be able to detect subtle changes. We compare changes over time of PMS symptom intensity and interference across three menstrual cycles. The *PMS symptom intensity score* is derived from the first 27 items of the German PMS symptom diary, and the *symptom interference score* is derived from the last three items. With regard to symptom intensity, two additional subscales are calculated by dividing the 27 symptom items into a psychological symptom intensity subscale and a physical symptom intensity subscale score (see secondary outcomes).[17]

Following Kues *et al*[18] and Janda *et al*,[16] we calculate separate scores for the luteal and follicular phase and consider the difference for our analysis. In more detail, we conservatively consider 7 days prior to menstruation onset as a time slot for the luteal phase and 7 days of the follicular phase, whereas the start of the follicular phase for each participant is calculated using the following formula: (individual cycle duration in days−14)/2=first day of the follicular phase. Subsequently, we calculate the difference score of the means for each menstrual cycle for the respective primary outcome of interest (ie, for PMS symptom intensity or interference). Intensity and interference are considered as marked if rated by participants with at least a 4 (very strong) on a scale from 0 (not at all) to 5 (extreme). Moreover, the item has to be rated with at least a four for at least

two consecutive days to be considered as marked. Further, symptoms only are considered as marked symptoms if the respective symptom is not marked in the follicular phase and if it is associated with marked interference in the luteal phase. If a participant has four or more symptoms marked in the luteal phase, also interference of only 1 day is accepted. We are also interested in the trajectory of the symptoms in the intervention groups across the two luteal phases after t0. Hence, we will compare if there are differences of group effects at cycle 2 (C2) and cycle 3 (C3) in contrast to cycle 1 (C1).

Moreover, we evaluate the effect of a comprehensive OLP treatment rationale, by providing one intervention group with a comprehensive treatment rationale in addition to intake instructions whereas the other intervention group only obtains intake instructions (see also the Trial treatment and study arms section).

### Secondary outcome measures

Secondary outcomes comprise the following:

### Physical and psychological symptom subscale of the German PMS Symptom Diary[16]

To test whether an OLP response occurs primarily on physical or on psychological symptoms, the above-mentioned German PMS Symptom Diary is used including the aforementioned calculation methods. The following scores are derived from the diary data of three menstrual cycles:

► Psychological symptom subscore: 14 items of the German PMS symptom diary are considered: depressed mood, hopelessness, feeling of worthlessness, feeling anxious, feeling tensed or being stressed out, irritability, sudden attacks of sadness, sudden crying, sensitivity to rejections, feeling angry, controversies with others, loss of interest in common activities, impaired concentration, feeling out of control or overwhelmed.

► Physical symptom subscore: The following seven items are considered: breast tenderness, breast sensitivity, breast swelling, headache, joint pains, muscle pain, feeling of a general body swelling, weight gain.

### PMS-Impact Questionnaire[19]

The PMS-Impact Questionnaire (PMS-I) measures the impact of PMS with 18 items and provides a total score and two subscales - a psychological impact and a functional impact scale. It was found to be a valid, reliable (Cronbach's alpha=0.90) and economic assessment tool of PMS impact.[16] The questionnaire differentiates between functional and psychological impact.[19] In comparison to the German PMS symptom diary, it covers more domains of daily life (eg, sexual experiences, physical exertion, stress coping) and is more specific for the detection of small changes of PMS impact. The PMS-I is assessed three times during the luteal phase, once during the prospective screening (L1) and twice during the intervention phase (L2 & L3).

### Short Form-12 Health Survey for health-related quality of life[20]

The Short Form-12 Health Survey (SF-12) measures health-related quality of life independent of disorders, comprising 12 items, providing a physical and mental component summary scale. Health-related quality of life is assessed since PMS symptoms have a high impact on women's quality of life.[21] Physical health is composed of the dimensions of physical functioning, physical role-functioning, bodily pain, general health; mental health is composed of the dimensions of vitality, social functioning, emotional role-functioning and mental health. The SF-12 is a validated and economic version of the SF36. For our purposes, we ask for ratings of the last week including today (premenstrual phase). The SF-12 is completed three times by participants: during the luteal phase of the prospective screening (L1) and twice during the intervention phase (L2 & L3).

### Hospital Anxiety and Depression Scale–German Version[22]

The Hospital Anxiety and Depression Scale–German Version (HADS-D) aims to assess anxiety and depression in individuals with somatic conditions and can be used for self-reported severity evaluation. It comprises two subscales (anxiety and depression) with seven items each and with a four-level answer format. The sum score indicates general psychological interference. We decided to use the HADS-D since it is a very sensitive tool for less severe cases of psychological disorders, which is usually the case for patients in a primarily somatic setting. We assess the HADS-D only at L1 during the first luteal phase before the start of the intervention.

### A scale for assessment of satisfaction in close relationships[23]

The scale for assessment of satisfaction in close relationships (ZIP) is the German version of the relationship assessment scale by Hendrick et al[24] and measures satisfaction in romantic relationships. The questionnaire entails seven items and had been shown to have high reliability and a clear unifactorial structure.[24] The ZIP is rated three times by participants during the luteal phases of the prospective screening (L1) and twice during the intervention phase (L2 & L3).

### Treatment adherence is measured by counting the remaining pills

Participants are asked to return the blister and pill boxes after the intervention.

► Expectancy of relief and desire for relief of premenstrual symptom severity are assessed at t0 by means of a Likert scale ranging from 1 to 6, like the primary outcome measure (German PMS Symptom Diary) and 3 weeks after intervention start.

### Complementary and Alternative Medicine Beliefs Inventory[25]

The Complementary and Alternative Medicine Beliefs Inventory serves to assess three different aspects of complementary medicine intervention beliefs with a 17-item questionnaire with satisfactory validity and reliability measures. Three dimensions comprise beliefs in natural treatments, participation in treatment and holistic health. Participants fill out the inventory post-treatment at t2.

## Item about placebo appraisal of the Questionnaire on Responders' Attitude Regarding Non-Specific Therapies[26]

At t2, participants rate the item about placebo appraisal online which encompasses the following three questions:
1. How would you define the term placebo?
2. A rating of the positivity or negativity of the term.
3. Do you believe that physical complaints can get better if you merely belief in the efficiency of a therapy?

## Patient–provider connection items of the Healing Encounters and Attitudes Lists–German Short Version[27]

The Healing Encounters and Attitudes Lists - German Short Version (HEAL-S) provides an item bank to measure nonspecific factors in treatment. We collect the patient–provider connection item at t2 to control for the influence of provider perception.

## Placebo intervention credibility is assessed online at t2 with the question

Did you believe that you received placebo pills during the study, that is, pills without any pharmacological content? The three answer options are:
1. Yes, I was sure that I received placebo pills.
2. I had doubts that I received placebo pills.
3. No, I did not believe that I received placebo pills.

Moreover, the OLP− group is asked whether they felt something was missing, whereas the OLP+ group is asked whether they found the provided rationale convincing and how it may have helped or confused them personally.

## Tertiary outcomes measures

Other, tertiary outcomes of interest are:
► Sociodemographic variables: age, family and relationship status, sexual orientation, citizenship, country of birth, mother tongue, educational degree, employment status, height, weight and number of births and aborts, if participant has a GP/gynaecologist.
► Retrospective PMS Questionnaire sum score.
► Mini-International Neuropsychiatric Interview.
► Pregnancy test.
► Medical and gynaecological history.
► Concomitant pharmacological or non-pharmacological interventions and changes in all groups.
► Qualitative answers regarding treatment and study experience.
► Occurrence of side-effects/adverse events.
► Number of participants who withdraw from study participation.
► Number of participants from the TAU group who attend an OLP intervention after study conclusion.
► Number of participants from the intervention groups who show interest to continue the OLP intervention after study conclusion.
► Suicidality.

## Adverse events and safety monitoring

Study participants will be asked about any side-effects, including also treatment-emergent adverse-events or serious adverse events at t1 and t2. Suicidality, which is specifically associated with premenstrual dysphoric disorders,[28] and generally found to be higher in the luteal (=premenstrual) phase,[29] will also be assessed at each study visit and if necessary, safety actions will be provided.

## SAMPLE SIZE AND STATISTICAL ANALYSIS OF DATA
### Data management

For data management, Microsoft Excel and LabKey, which is hosted by the University of Basel, is used. Study data are collected at the web server-based software Lime-Survey. Only authorised study team members are able to view and export data. Entries and actions are marked with the initials of the respective study member.

### Sample size

A calculation with a power analysis on the basis of an F-test and an analysis of covariance for three groups, using the statistical software G*Power, indicated a total sample size of 206 participants needed for a power of 0.9 and a total sample size of 158 participants for a power of 0.8 to detect a medium effect size of f=0.25 (equals d=0.5) with an alpha-level of 0.05.

Since we will adopt a multilevel approach which enables more precise estimations, we also conducted a power analysis on a repeated-measures analysis of variance, using G*Power, whereby we are interested in the interaction between time (menstrual cycles, 3 levels) and study groups (3 levels). If the correlation for the outcome among the different time points (1–3 cycles) is assumed to be 0.5, then based on an alpha-level of 0.05, a power of 0.8 and a medium effect size of f=0.25, the required samples size is 36 (12 per group). Accordingly, for the main effect of menstrual cycle, we would need 30 participants (10 per group), and for the main effect of study group we would need 108 participants (36 per group). On this basis, we chose a conservative total sample size of 150 participants (50 per group). It is worth noting that most OLP studies reported effect sizes that are generally higher than d=0.5 (eg, chronic low back pain: d=0.77[1]; irritable bowel syndrome: d=0.79).[2] However, in another OLP study for patients suffering from major depressive disorder,[5] the authors reported an effect size of d=0.54 while stating that the study was limited by low statistical power (ie, the findings did not support the hypothesis that OLPs are effective in depression). Since PMS is not only characterised by physical complaints, yet also by psychological symptoms such as depressive mood, and since there is no specific research on OLP effects and effect sizes in PMS, we considered a conservative medium effect size as more appropriate. This resulted in a more conservative sample size.

### Statistical analysis

For the testing of the primary hypotheses, the daily data from three menstrual cycles of a symptom diary is examined. Because of the hierarchical structure of the data (menstrual cycles as level-1 are nested within

participants as level-2) and given our aim to predict post-interventional changes between menstrual cycles as the primary outcome, a multilevel modelling approach will be adopted. Multilevel models take care of the interdependence of the hierarchical data that arises due to the fact that observations within the same individual are typically more similar to each other compared to observations between individuals.[30 31] Here, we will analyse temporal changes across menstrual cycles of symptom intensity and interference among the different study groups. To compare a TAU group with two intervention groups and to evaluate the effect of the applied treatment rationale, the following linear a priori contrast will be tested for the factor study group: TAU<OLP–<OLP+. Thus, based on published findings,[6] we expect that the effect of OLP– lies between that of TAU and OLP+. Out statistical model contains menstrual cycle as within-subject factor with three levels (C1, C2 and C3), study group as between-subject factor with three levels (TAU, OLP+ and OLP–), and the interaction between the two factors. In line with recommendations by Fitzmaurice *et al*,[32] we will treat baseline values (ie, values at C1) as a part of the outcome vector and assume that the means among study groups are equal at baseline since participants are randomised. The model contains the two main effects study group contrast (defined above) and menstrual cycle (temporal effect) plus the interaction between the two. We will be primarily interested in the interaction term as it tests for the differential temporal change between study groups in order as defined in the linear contrast.

Analyses of secondary outcomes are based on the same multilevel model as used for primary outcomes, but will be considered exploratively, thereby not correcting for multiple tests.[33]

## Monitoring

The study is monitored for quality and regulatory adherence. The monitor, who is not involved in our study and approved by the local ethical committee, verifies the qualification of the investigators and study team members, and monitors sound and appropriate documentation.

## Ethics and dissemination

The present protocol and applied informed consent forms were approved with regard to their content and compliance with ethical regulations by the Ethics Committee Northwest and Central Switzerland. The study is carried out with principles enunciated in the current version of the Declaration of Helsinki[34] and the guidelines of Good Clinical Practice issued by the International Conference on Harmonization.[35]

Women interested in study participation receive a participant information sheet and consent form, describing the study and providing sufficient information for an informed decision about participation and data confidentiality and detailed oral information is provided. The results of the planned analyses will be published in a peer-reviewed journal.

## DISCUSSION

Given the huge burden of women suffering from PMS,[11 36] it is necessary to optimise treatment options and to enhance the understanding of intra-interventional processes in PMS treatment. Moreover, interventions - including medical or psychotherapeutic treatment - can generally benefit from an increased comprehension of treatment context-factors and related meaning responses[37 38] involved in ailment amelioration. We therefore address the investigation of PMS-interventional context-factors and OLP effects and mechanisms with our study and strive to contribute to further knowledge in both areas. Previous randomised controlled trials of PMS interventions indicate high placebo responses.[7] The presented trial is essential to test whether an OLP response can also be elicited in PMS and whether an amelioration of symptoms occurs primarily on a somatic or psychological level. Additionally, we assess the importance of a comprehensive treatment rationale in a clinical population independent of a specific intervention.

So far, there have been different aspects of previous OLP studies which have been critically evaluated in a meta-analysis of clinical OLP studies.[39] For instance, a lack of an adequate control group in OLP trials has been pointed out. To address this issue, the TAU control group of our trial has the same amount of contact with the study staff as the two intervention groups. Furthermore, we added an OLP– group to test the effect of a comprehensive OLP treatment rationale in a clinical population. Hence, we aim to assess whether placebo pills alone are sufficient or whether a comprehensive treatment rationale is vital to elicit an OLP response and, thus, strive to distinguish OLP effects from positive framing.[39]

Moreover, to tackle previous critique of potential researcher bias and lack of blinding,[39] any interactions with participants regarding the study proceedings, including data collection, are conducted by study team members who do not provide the intervention and who are not informed about study group allocation at the study visit. The intervention is provided by one person only (AFN) who is not involved in any interactions related to data collection. Furthermore, we decided to not provide the treatment rationale to all participants, primarily to test the effect of the treatment rationale and to avoid any experiences of disappointment in the TAU control group. A further advantage of our study is the large planned sample of 150 participants, tackling a further previously mentioned critique point.[39]

Because we are interested in understanding mechanisms of OLP responses, we assess an array of different questionnaires including attitudes and experiences of participation in the different study groups. In this sense, we aim to address different aspects of former criticism with our study design and to enhance knowledge about OLP effects.

To sum up, a positive OLP response in PMS would form the basis for future studies investing how to successfully harness placebo effects, in an ethical fashion, in clinical

practice and to optimise PMS interventions that often entail side-effects.

**Contributors** All of the following authors contributed significantly to the conception of the presented study and to the manuscript: AFN, JG, IK, JK, AM, CL.

**Funding** This study is supported by the Swiss National Science Foundation (JG), grant number: 325130_170117, under the name "Taking the placebo further: open placebo". Cosima Locher, PhD, received funding for this project from the Swiss National Science Foundation (SNSF): P400PS_180730 (Title: Overcoming Classificatory and Methodological Hurdles to Improve Treatment of Chronic Primary Pain: A Network Meta-Analytic Approach).

**Competing interests** None declared.

**Patient consent for publication** Not required.

**Ethics approval** This study has been approved by the EKNZ Ethical Committee (ID 2017-02186; Open-label placebo treatment of women with premenstrual syndrome: A randomised controlled trial).

**Provenance and peer review** Not commissioned; externally peer reviewed.

**ORCID iDs**
Antje Frey Nascimento http://orcid.org/0000-0001-7907-0439
Cosima Locher http://orcid.org/0000-0002-9660-0590

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
