## [Reviewer comments · BMJ Open]

ARTICLE DETAILS

TITLE (PROVISIONAL)	OPEN-LABEL PLACEBO TREATMENT OF WOMEN WITH PREMENSTRUAL SYNDROME: STUDY PROTOCOL OF A RANDOMIZED CONTROLLED TRIAL
AUTHORS	Frey Nascimento, Antje; Gaab, Jens; Kirsch, Irving; Kossowsky, Joe; Meyer, Andrea; Locher, Cosima

VERSION 1 – REVIEW

REVIEWER	Takashi Takeda Division of Women's Health, Research Institute of Traditional Asian Medicine, Kindai University, Japan
REVIEW RETURNED	07-Sep-2019

GENERAL COMMENTS	1) In section of abstract methods, please explain statistical tests that will be used.2) What is the primary outcome measure? You should explain in section of abstract methods.3) Methods and analysis, study design, you should show the approved number of the ethical committee.4) Same section, you should show the trial schedule as a table.5) In Figure 1, what is the shaded area?6) Trial treatment and study arm, please explain about the contents of placebo.7) Table 1 shows that two intervention groups obtain placebo pills, two pills per day for six weeks, but there is no description in methods and analysis section.8) Why did you choose six weeks intervention? You must show sufficient cause for selecting such short intervention period.9) When do they start pills? From Day 1?
---

REVIEWER	Michael Schaefer Medical School Berlin
REVIEW RETURNED	03-Oct-2019

GENERAL COMMENTS	The authors present a study protocol on OLPs as a treatment of premenstrual syndrome. They include two OLP conditions, one with a scientific treatment rationale and one without, as well as a TAU group. The study protocol is well written. I have only few points. 1. Since one of the main interesting parts of this protocol is the implementation of two OLP-conditions, this important fact should be mentioned in the abstract.
---

	2. In the introduction the authors claim that PMS “offers the opportunity to investigate OLP effects on both somatic and psychological symptoms”. This should be explained more in detail. 3. At the end of the introduction the authors hypothesize that OLP treatment with a rationale will improve more than OLP without treatment rationale. However, in table 1 they assume “a different OLP response” here. This should be clarified and the hypothesis justified (e.g., with the previous work of the authors). 4. How will the subjects be allocated to the study groups? 5. Figure 1: Why are there empty fields? Is there a problem with the quality of Fig. 1? 6. Please describe more in detail the crucial part of what information the OLP + group receives. For example, what kind of information do the participants get about “why placebos work”? Perhaps it would be good to provide the original information here or in a table. Are the information given in a written way or orally? 7. Are the placebos given in a bottle or in a box? What is exactly in it? How do they look like? 8. It is very interesting that the TAU group is not provided with the treatment rationale, in order to avoid feelings of disappointment. But what kind of information does the TAU group receive exactly? Is it the same as the OPL group without treatment rationale? Or do they get even less information? What is inside the study information, which every participant at the beginning receives? Since the call of this study describes “a novel efficacy study of side-effect free intervention”, isn’t possible that the TAU group may still feel disappointed not to get any treatment at all? 9. Given that the lack of information on the treatment rationale for the TAU group seems novel; this fact should be mentioned in the manuscript earlier and more explicitly.
--	---

VERSION 1 – AUTHOR RESPONSE

Reviewer 1

- 1) In section of abstract methods, please explain statistical tests that will be used.
 - o We thank reviewer 1 for the appreciated feedback and suggestions. Regarding the first comment, we added the statistical procedure that will be used for data analysis in the Abstract’s sub-section Methods and analysis:

“Multilevel modelling will be used for data analyses.”

- 2) What is the primary outcome measure? You should explain in section of abstract methods.
 - o The primary measures are PMS symptom intensity and interference which is continuously assessed by means of the German PMS symptom diary, which also serves for the prospective PMS screening (i.e., study eligibility). To clarify this point, we adapted the respective part in the Abstract’s sub-section Methods and analysis:

“Besides the primary outcome PMS symptom intensity and interference, an array of further variables is assessed.”

- 3) Methods and analysis, study design, you should show the approved number of the ethical committee.
 - o We added the approved number of planned study participants in the Methods and analysis and Study design section:

“The first participant was enrolled and randomized in August 2018 and the study is expected to be concluded by spring 2020 with the planned inclusion of 150 study participants.”

o Also, the full name of the local ethical committee and the study ID were added to the aforementioned section:

“The study is being conducted at the Division of Clinical Psychology and Psychotherapy at the Faculty for Psychology of the University of Basel and was approved by the local ethical committee Ethics Committee for Northwest/Central Switzerland EKNZ (ID 2017-02186).”

• 4) Same section, you should show the trial schedule as a table.

o We added a schedule table of the trial in the section Methods and analysis and Study design (see page 11).

• 5) In Figure 1, what is the shaded area?

o Unfortunately, there was a problem with the display of Figure 1 and the words in the shaded area were not displayed. Therefore, we modified the figure and uploaded a new Figure 1. Now all words should be good readable.

• 6) Trial treatment and study arm, please explain about the contents of placebo.

o We administer original German placebos which are available in pharmacies in Germany without prescription. The pills look like dragées and are pink; the main ingredients are lactose and magnesium. Information about appearance, labeling and content of the placebo pills has also been added to the sub-section Trial treatment and study arm:

“As investigational product, pink oral placebo pills - “P-Dragees rosa Lichtenstein” of the German brand Winthrop - are administered, which are certified placebo pills without any active ingredient and with validated quality. Each pill contains the following substances: lactose monohydrate; magnesium stearate (Ph. Eur.); microcrystalline cellulose; highly dispersed silicon dioxide; white clay, magrogol glycerolhydroxy stearate (Ph. Eur.); Arabic gum; montanglycol wax; povidone (K 25); talcum; titanium dioxide (E 171); erythrosine; aluminium salt (E 127); calcium carbonate; sucrose; glucose syrup; maize starch; macrogol 6000.”

• 7) Table 1 shows that two intervention groups obtain placebo pills, two pills per day for six weeks, but there is no description in methods and analysis section.

o Many thanks for this question. This detail about placebo pill intake can be found also in the Methods and analysis sub-section Trial treatment and study arms:

“The intervention consists of an administration of open-label placebo pills including the instruction to take two pills a day for six weeks.”

• 8) Why did you choose six weeks intervention? You must show sufficient cause for selecting such short intervention period.

o Previous open-label placebo interventions were in comparison even shorter (see e.g., Kaptchuk et al., 2010 = 3 weeks; Carvalho et al., 2016 = 3 weeks; Schaefer et al., 2016 = 2 weeks). Due to the cyclicity of PMS symptoms, we decided to provide the intervention for a longer period of time to encompass at least one whole menstrual cycle and two PMS phases. However, given that participants have to conclude daily surveys and hence, study participation entails a considerable effort load for participants, we decided to not exceed 6 weeks of pill intake. This is also in line with ethical principles of good clinical practice, regarding cost-effectiveness-balance. We also added an explanation in the Trial treatment and study arm sub-section:

“Due to the cyclicity of PMS symptoms, we decided to provide the intervention for a longer period of time in contrast to previous OLP studies (1-3) to encompass at least one whole menstrual cycle.”

References:

1 Kaptchuk, T. J., Friedlander, E., Kelley, J. M., Sanchez, M. N., Kokkotou, E., Singer, J. P., Kowalczykowski, M., Miller, F. G., Kirsch, I. & Lembo, A. J. (2010). Placebos without deception: a randomized controlled trial in irritable bowel syndrome. *PloS one*, 5(12), e15591.

2 Carvalho, C., Caetano, J. M., Cunha, L., Rebouta, P., Kaptchuk, T. J., & Kirsch, I. (2016). Open-label placebo treatment in chronic low back pain: a randomized controlled trial. *Pain*, 157(12), 2766.

3 Schaefer, M., Harke, R., & Denke, C. (2016). Open-label placebos improve symptoms in allergic rhinitis: A randomized controlled trial.

• 9) When do they start pills? From Day 1?

o We agree that this point should be clearer presented in the study protocol. The pill intake in both OLP intervention groups starts at the very day after the study visit (t0) and goes on for 6 weeks. The study visit (t0) occurs during the first 14 days of the menstrual cycle of each participant, i.e., in the follicular phase. All participants obtain the same amount of placebo pills and intake is monitored by a daily question in the symptom diary surveys as well as at the end by pill count. We added this clarification also in the manuscript in the Methods and analysis sub-section Trial treatment and study arms: "The pill intake in both OLP intervention groups starts at the very day after the study visit (t0). The study visit (t0) occurs during the first 14 days of the menstrual cycle of each participant, i.e., in the follicular phase."

Reviewer 2

• The authors present a study protocol on OLPs as a treatment of premenstrual syndrome. They include two OLP conditions, one with a scientific treatment rationale and one without, as well as a TAU group. The study protocol is well written. I have only few points.

o We thank reviewer 2 for his very helpful feedback.

• 1. Since one of the main interesting parts of this protocol is the implementation of two OLP-conditions, this important fact should be mentioned in the abstract.

o We added the fact of having two OLP conditions also to the Abstract sub-section Methods and analysis:

"To test the efficacy of OLPs in women suffering from PMS, a clinical randomised controlled trial including two OLP study groups (with and without treatment rationale) was designed to investigate on the effect on PMS."

• 2. In the introduction the authors claim that PMS "offers the opportunity to investigate OLP effects on both somatic and psychological symptoms". This should be explained more in detail.

o We added further information in the Introduction and Rationale part regarding the assessment of somatic as well as of psychological symptoms, which are both assessed by sub-scales of the German PMS symptom diary:

"Furthermore, PMS offers the opportunity to investigate OLP effects on both somatic and psychological symptoms since symptom diaries like the German PMS symptom diary inquire about physical (e.g., breast tenderness, headaches, joint and muscle pain) as well as psychological symptoms (e.g., depressed mood, hopelessness, irritability)."

• 3. At the end of the introduction the authors hypothesize that OLP treatment with a rationale will improve more than OLP without treatment rationale. However, in table 1 they assume "a different OLP response" here. This should be clarified and the hypothesis justified (e.g., with the previous work of the authors).

o Since we expect an enhanced OLP response in the OLP group with treatment rationale in comparison to the OLP group without treatment rationale, we modified the research question in accordance with our stated hypothesis in the introduction in Table 1, section Research questions:

“Do participants who receive a comprehensive OLP treatment rationale show a greater OLP response in comparison to participants who do not receive any treatment rationale over time?”

- 4. How will the subjects be allocated to the study groups?

- o Subjects are randomly allocated to the study groups at the t0 study visit, which marks the start of the intervention phase. With their appearance at the study site for the start of the intervention phase, they obtain their individual randomization number which allocates them to one of the three study groups randomly. For the creation of the random allocation sequence we used the built-in random number generator of Microsoft Excel®. Women are informed about their respective allocation of study groups after a pregnancy test and the extensive screening of mental disorders at the study site, if they fulfill all eligibility criteria. We also added the following in the sub-section Trial treatment and study arms of the section Methods and analysis:

“Following recruitment and verification of study eligibility - including a prospective PMS screening for one menstrual cycle - qualified women are randomly allocated by means of a random allocation sequence using the built-in random number generator in Microsoft Excel® to one of the three study groups at the study visit (t0) and also are informed before any intervention, respectively the control contact starts: 1.) a TAU group; 2.) an OLP+ group, and 3.) an OLP- group (see also Figure 2).”

- 5. Figure 1: Why are there empty fields? Is there a problem with the quality of Fig. 1?

- o Unfortunately, there was a problem with the display of Figure 1 and the words in the shaded area were not displayed. Therefore, we modified the figure and uploaded a new Figure 1. Now all words should be good readable.

- 6. Please describe more in detail the crucial part of what information the OLP + group receives. For example, what kind of information do the participants get about “why placebos work”? Perhaps it would be good to provide the original information here or in a table. Are the information given in a written way or orally?

- o Information is given orally. Afterwards, participants receive an information pamphlet which has to remain at the study site to avoid someone in the ‘OLP without treatment rationale’ group from obtaining the information. To this end, participants in all groups are asked to not provide any content information they obtain during the study visit to people who are already taking part, could potentially take part, or who might know someone who could take part in the study. We uploaded our OLP treatment rationale as a supplement.

- 7. Are the placebos given in a bottle or in a box? What is exactly in it? How do they look like?

- o The placebos are distributed in their original packaging which is a paper box. We administer original German placebos which are available in pharmacies in Germany without prescription. The pills look like dragées and are pink and the main ingredients are lactose and magnesium. Information about appearance, labeling and content of the placebo pills has also been added to the section Trial treatment and study arm:

“As investigational product, pink oral placebo pills - “P-Dragees rosa Lichtenstein” of the German brand Winthrop - are administered. They are certified placebo pills without any active ingredient and with validated quality. Each pill contains the following substances: lactose monohydrate; magnesium stearate (Ph. Eur.); microcrystalline cellulose; highly dispersed silicon dioxide; white clay, magrogol glycerolhydroxy stearate (Ph. Eur.); Arabic gum; montanglycol wax; povidone (K 25); talcum; titanium dioxide (E 171); erythrosine; aluminium salt (E 127); calcium carbonate; sucrose; glucose syrup; maize starch; macrogol 6000.”

- 8. It is very interesting that the TAU group is not provided with the treatment rationale, in order to avoid feelings of disappointment. But what kind of information does the TAU group receive exactly? Is it the same as the OLP group without treatment rationale? Or do they get even less information?

o We designed the study in that way that all participants receive the same amount and quality of contacts and whenever possible the same verbal script. Hence, the verbal script is the same for all three study groups in the study visit, differing only in that the TAU group does not receive any information about the intervention, and the OLP- group receives instructions concerning pill intake, but not the rationale. However, at the beginning of the intervention phase, each group is told why participation in this group is particularly valuable for our study team and research. Thus, the TAU group is told that this specific group is very valuable because only with their help it is possible to evaluate the effectiveness of our intervention by controlling for the natural course of disease. The OLP- group, in turn, is told that this specific group is important because it would be the aim of the study to find out if and how different interventions of PMS work and thus, the OLP- group (for participants labeled as 'placebo group') is important for the study as well as for research on effective interventions of PMS. Again, the groups are told at the t1 phone contact why the respective study group is very valuable to the study, and TAU participants are again reminded that they will be asked at the t2 phone contact if they desire the treatment after their study conclusion. Furthermore, at t0 and t1, all groups are asked and reminded to continue with the daily completion of the symptom diary. In this way we hope to decrease study drop out and missing data. To make our procedure clearer in the study protocol, we added the following to the Methods and Analysis' sub-section Trial treatment and study arms: 1) The TAU group:

"However, participants of this group do not receive any intervention nor any information concerning the intervention. However, they are told why the control group is valuable and essential for the trial and that they will receive the intervention after study conclusion if desired."

- What is inside the study information, which every participant at the beginning receives? Since the call of this study describes "a novel efficacy study of side-effect free intervention", isn't possible that the TAU group may still feel disappointed not to get any treatment at all?

o Before written study information obtainment, interested women receive an email with information, in which we mention that the novel treatment method entails among other aspects also the intake of placebo pills. This is intended to be honest but not too explicit, so as to avoid a strong bias by only attracting women who would be open to a placebo intervention. Also, in the study information form the administration of placebo pills for both groups - "the placebo group" (=OLP-) and "the treatment group" (=OLP+) as they are labeled for study participants - is mentioned. The study participation consent form was also uploaded as a supplementary file in German as well as an English translation.

o Regarding the question of disappointment in the TAU group, we want to emphasize that all participants of the TAU group are reminded during study that they will have the opportunity to obtain the treatment after their study participation.

- 9. Given that the lack of information on the treatment rationale for the TAU group seems novel; this fact should be mentioned in the manuscript earlier and more explicitly.

o We added this fact in the manuscript in the section of the Introduction and Rationale:

"Second, we assume that women receiving an OLP treatment without a treatment rationale will also show a higher decrease in PMS symptom intensity and interference in comparison to the treatment as usual group (TAU), which is not given any of the treatment-specific information that the OLP groups receive nor the rationale for the OLP treatment."

VERSION 2 – REVIEW

REVIEWER	Takashi Taakeda Kindai University, Japan
REVIEW RETURNED	30-Nov-2019

GENERAL COMMENTS	I am pleased with authors' answers to mine and other reviewers' comments. The authors responded to comments appropriately. But Figure1 is still unreadable.
--

REVIEWER	Michael Schaefer Medical School Berlin Germany
REVIEW RETURNED	27-Nov-2019

GENERAL COMMENTS	The authors addressed all my points. I have no further objections.
--